# A study of UK household wealth through empirical analysis and a non-linear Kesten process

Samuel Forbes[1☯], Stefan Grosskinsky[2☯]*

**1** Mathematics Institute, University of Warwick, Coventry, United Kingdom, **2** Institute of Mathematics, University of Augsburg, Augsburg, Germany

☯ These authors contributed equally to this work.
* Stefan.Grosskinsy@math.uni-augsburg.de

**Data Availability Statement:** All data are available at https://github.com/saf92/PLOS-ONE-Kesten. Sources of publicly available data are cited in the repository as well as the article.

## Abstract

We study the wealth distribution of UK households through a detailed analysis of data from wealth surveys and rich lists, and propose a non-linear Kesten process to model the dynamics of household wealth. The main features of our model are that we focus on wealth growth and disregard exchange, and that the rate of return on wealth is increasing with wealth. The linear case with wealth-independent return rate has been well studied, leading to a log-normal wealth distribution in the long time limit which is essentially independent of initial conditions. We find through theoretical analysis and simulations that the non-linearity in our model leads to more realistic power-law tails, and can explain an apparent two-tailed structure in the empirical wealth distribution of the UK and other countries. Other realistic features of our model include an increase in inequality over time, and a stronger dependence on initial conditions compared to linear models.

## 1 Introduction

The dynamics of wealth and income inequality is a subject of increasing research interest and public debate, encapsulated by major works such as Piketty's 'Capital in the 21st Century' [1]. The recent COVID-19 pandemic has added to the debate on inequality as some of the very richest, particularly in the tech industry, have gained large quantities of wealth whilst many 'ordinary' households have faced redundancies and reliance on government benefits [2]. Data on standard inequality measures, such as the Gini coefficient as well as wealth or income shares, clearly indicate that inequality has increased since the 1980s in many areas of the world [3]. Potential contributing factors include globalisation, financialisation, decreased taxes, increased tax evasion and avoidance, increased inheritance and domination of the technological sector [1, 3, 4]. In this paper we summarise these multitude of factors into an idealised growth model for household wealth, dominated by one simple effect: that the wealthier you are, the higher your rate of return (ROR), i.e. the return on wealth you are likely to receive grows superlinearly with wealth. We refer to this type of reinforcement dynamics in our discrete time model as a non-linear Kesten process, which is a generalisation of the work on linear

**Funding:** S.F., EP/L015374/1, Engineering and Physical Sciences Research Council, https://epsrc.ukri.org/ The funders had no role in study design, data collection and analysis, decision to publish, or preparation of the manuscript.

**Competing interests:** The authors have declared that no competing interests exist.

reinforcement initiated by Kesten [5]. The increasing dependence of RORs on wealth has been confirmed in recent studies [6–8], and we present further empirical evidence for the UK.

Our model uses an agent-based approach, which describes the wealth of individual households as a function of time. The dynamics of individual agents is kept as simple as possible (in our case they evolve independently) and the goal is to predict the collective behaviour of the system via statistical properties of the ensemble of agents. Stochastic agent-based models with multiplicative noise applied to income and wealth dynamics have a long history in economics, with an early major publication in 1953 by Champernowne [9], and since then have been applied extensively and are summarised in several reviews, see for example [10, 11]. These models have been used as they exhibit power-law tails, which is a key feature of both income and wealth distributions. Research in the field of Econophysics has focused mostly on exchange of money or wealth, in analogy to energy transfer in models of statistical mechanics (see e.g. [12, 13] for an overview). It has been found also in this context, that additive noise leads to Boltzmann-Gibbs type distributions with exponential tails, and heavy tails can result from multiplicative noise or disorder [14]. The focus on pure exchange dynamics has been recognised as unrealistic to model wealth (see [12] page 13), but only very few studies consider both exchange and growth. In [15, 16] the authors study growth dynamics of wealth with a global redistribution dynamics, inducing a weak mean-field type interaction between agents. In our model we disregard wealth exchange between households and focus entirely on growth dynamics. This is of course a simplification, but in our view and in line with previous studies mentioned above, growth is clearly the dominant aspect of wealth dynamics for most households, and on average nominal wealth has been growing in an exponential fashion since at least the industrial revolution [17].

Fig 1 shows the tail of the household wealth distribution for the UK from recent wealth and asset survey (WAS) data [18] and rich lists [19, 20]. We see here the presence of two power laws in the upper tail: one for the richest in the survey with exponent around 2, and one for the richest in society found in the rich lists with exponent around 1. Such a change in power-law exponent has been observed for other countries [21], and is often argued to be a sampling artefact from survey bias in the data [21, 22]. However, due to the particular strength of the effect we believe that the two-tailed structure is a genuine feature of the data. From previous studies [23] linear Kesten processes are known to lead to asymptotic log-normal distributions of wealth. Our non-linear model produces a power-law tail from various generic initial conditions, and in the long run also a two-tailed structure due to a crossover phenomenon resulting from the non-linearity, which we will explain in detail.

We also find that our model has a strong dependence on initial conditions, corresponding to the idea of a low social mobility [24]. It is particularly suitable to describe wealth dynamics since the 1980s, when deregulation of financial markets started to facilitate increasing rates of return for assets typically held by wealthier agents [25], providing increased access to credit and investment opportunities. During the 2007–2008 financial crisis, shortage of available credit temporarily also affected wealth growth for households [26]. But after a relatively short period of adaption and in spite of declining interest rates [27], prices of e.g. housing and financial assets are again increasing at close to pre-crisis levels [28], so the main premise of our model remains valid. While an important macroeconomic question, the mechanisms behind wealth growth are not part of our discussion and we focus on the distribution of wealth among households. Throughout this paper we only model positive wealth, while appreciating that a significant fraction (above 10% [18]) of the UK population has negative wealth, i.e. is in debt. This requires additional modelling and the dynamics we propose do not apply in this case.

Wealth can be defined as assets minus liabilities [29] and is usually measured in a particular currency, GBP in our case. It can be interpreted as the balance sheet of a household, and

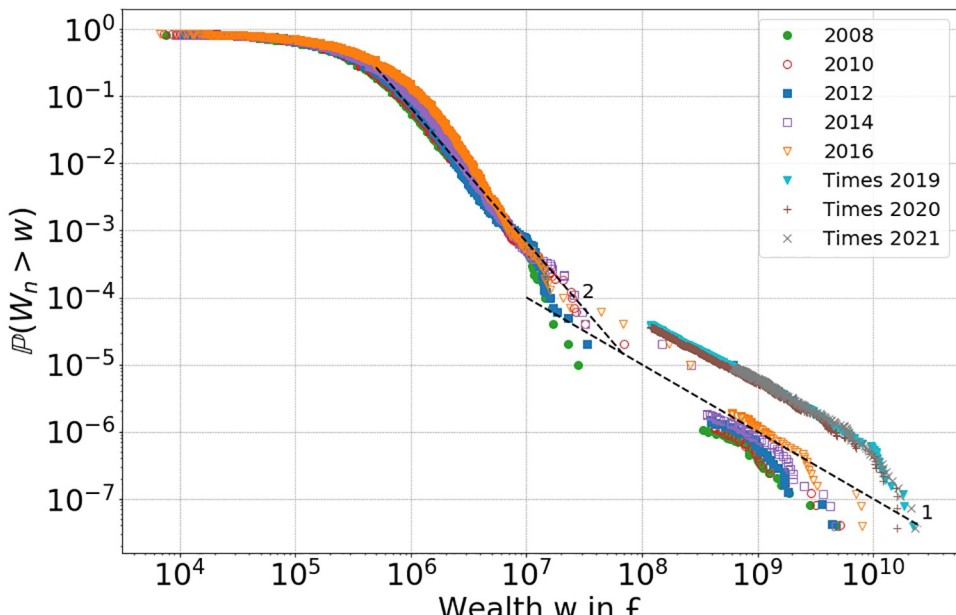

**Fig 1. Empirical tail distribution of positive UK household wealth for five consecutive time periods 2008, 2010, 2012, 2014, 2016 from the WAS survey [18], together with Forbes rich list data on billionaires [19] (same colour code), and UK Times rich list data from 2019, 2020 (see Section SI.2.1 in S1 Appendix).** Dashed lines indicate power-law tails with exponents 2 and 1 for comparison.

therefore only assets that can be assigned a monetary value contribute, excluding e.g. health or education of members of a household. We also note that wealth is a stock of value unlike income, which represents a flow of value over time. The WAS categorises wealth into four components: physical, financial, property and pension [30]. In our model we assume that wealth increases on average due to two mechanisms: multiplicative growth due to returns on current wealth, and additive residual savings such as excess salary that is not spent on living costs and other expenses which do not contribute to the balance sheet of the household. In general, wealthier agents can diversify their assets, including riskier strategies with higher average returns [7, 8]. Different composition of wealth in different wealth deciles is provided in the WAS [18] and summarised in Section SI.2.4 in S1 Appendix.

## 2 Model

We consider independent agents (representing households), whose wealth at discrete time $n \in \{0, 1, 2, \ldots\}$ (representing years) is denoted by $W_n > 0$. As explained in the introduction, we focus on wealth growth rather than exchange, and model the dynamics of positive wealth only, keeping track of bankruptcy events after which we reset the wealth value of the agent (see Section 4 for details). We assume that the wealth of an agent over the time period $n$ to $n + 1$ changes via two mechanisms: **returns on existing wealth**, where $R_{n+1} \in \mathbb{R}$ denotes the corresponding rate of return (ROR), and **residual savings** $S_{n+1} \geq 0$, resulting for example from excess earnings which are independent of the current wealth of an agent (see Section 3.3 for details). This leads to the recursion

$$W_{n+1} = W_n(1 + R_{n+1}) + S_{n+1} \quad \text{with initial condition} \quad W_0 > 0. \tag{1}$$

Here the RORs $R_n$ and residual savings $S_n$ are independent random variables. It is commonly accepted that RORs depend monotonically on wealth [6–8], and we assume the

following power-law form,

$$R_{n+1} = \alpha_{n+1} W_n^{\gamma-1} \quad \text{for some } \gamma \geq 1, \tag{2}$$

where $\alpha_n \in \mathbb{R}$ are i.i.d. random variables from some fixed probability distribution, and with small probability can also take negative values. The very simple choice (2) is consistent with empirical data for the UK presented in Section 3.1. We are not claiming that this is the best or most detailed model for RORs, which have been observed in some cases to exhibit an intermediate plateau rather than a strict increase as a function of $W_n$ (see e.g. Fig 2 in [8]). But our aim here is to capture the most essential features in a simple model that can also be analysed mathematically, and it is of course possible in simulations to replace (2) by different functions. We find that a non-central $t$ distribution (see Section SI.2.6 in S1 Appendix for details) provides a good match with data for $\alpha_n$, which is discussed in Section 3, Fig 4.

Substituting (2) in (1) gives the recursion

$$W_{n+1} = W_n + \alpha_{n+1} W_n^\gamma + S_{n+1}. \tag{3}$$

With $\gamma > 1$ we refer to (3) as a **non-linear Kesten process**. We now summarise theoretical results of (3) for different values of $\gamma$.

$\gamma = 1$. In this case $R_n = \alpha_n$ and $W_{n+1} = (1 + \alpha_{n+1})W_n + S_{n+1}$. The stationary version of this linear model has been introduced and studied by Kesten [5], and the non-stationary asymptotic growth case is more recently discussed in [23]. It is easy to see that the asymptotic behaviour of $W_n$ is dominated by the exponential $e^{n\log|1+\alpha_n|}$, and we present details on the analysis of both cases in Section SI.1.2 in S1 Appendix. In the stationary case with $\mu := \mathbb{E}[\log|1 + R_n|] < 0$, the model is known to exhibit power-law tails in the limiting distribution, but for wealth dynamics the non-stationary case of asymptotic growth is most relevant, which occurs for $\mu > 0$. Following results in [23], the asymptotics is given by a **log-normal distribution** such that to leading exponential order

$$W_n \asymp W_0 \exp(\mu n + \sqrt{n v^2} Z) \quad \text{as } n \to \infty, \tag{4}$$

where $v^2 := \text{Var}[\log|1 + R_n|]$ and $Z \sim \mathcal{N}(0, 1)$ is a standard Gaussian. Here the symbol $\asymp$ means that $W_n = W_0 \exp(\mu n + \sqrt{n v^2} Z + o(\sqrt{n}))$ as $n \to \infty$, where $o(a_n)/a_n \to 0$ for all sequences $(a_n : n \in \mathbb{N})$ with $a_n \to \infty$. The rigorous version of this result is subject to further reasonable and mild regularity assumptions on the distributions of parameters (see Theorem 2 (i) in [23]), and the leading order behaviour is independent of the residual savings $S_n$. Since (3) is linear in $W_n$, the model also has a natural scale invariance for the units of wealth (see discussion in [16]), and the initial condition $W_0$ enters (4) as a simple multiplicative constant.

$\gamma > 1$. To our knowledge the non-linear model has not been studied before. Details are given in Section SI.1.3 in S1 Appendix, where we find asymptotic super-exponential growth to leading order,

$$W_n \asymp \left(W_0 e^D\right)^{\gamma^n} \quad \text{as } n \to \infty, \tag{5}$$

where $D$ is given by a convergent series depending on the distribution of $\alpha_n$ and the initial behaviour of the process. Again, we focus on the non-stationary case with $W_0 e^D > 1$. In contrast to the linear case, we see that the asymptotics depend in a strong, non-linear way on the initial conditions and early dynamics of the process. Therefore there is no central limit theorem on the logarithmic scale that leads to (4), and we are not able to predict the asymptotic scaling distribution of $W_n$. But numerical results presented in Section 4 show that the model exhibits power-law tails with realistic shapes on relevant time scales.

For realistic initial conditions and parameters the dynamics follows initially an exponential growth regime, and super-exponential growth sets in when the dominant term in yearly gains in Eq (3) changes from $W_n$ to $\alpha_{n+1} W_n^\gamma$ (additive residual savings again do not influence the asymptotic behaviour). This means that the returns from wealth in a single year become of the same order or higher than current wealth, which happens for values around

$$W_n \approx \alpha_{n+1}^{-1/(\gamma-1)}. \tag{6}$$

Billionaire return data in Fig 2 below indeed confirm that RORs of around 100% or more can be achieved. From numerical results in Section 4 we see that this crossover leads to a two-tailed structure of the distribution of $W_n$ similar to what we see in the data in Fig 1, and we think this feature of the model provides a promising explanation for this effect. Since we find in the next section that $\gamma$ is close to 1, (6) is very sensitive to the value of the random variable $\alpha_{n+1}$ (which is raised to a large power), leading to a broad crossover region. While this cross-over is a realistic feature seen in data from the UK and other countries ([21], but notably not in the USA, see online Appendix of [21]), the non-linearity also implies that the model is not scale invariant and coefficients will depend on the currency unit.

We further find empirically that $\alpha_n$ is mostly positive with a heavy tail, but negative values are possible, see Fig 4 of Section 3.2, and thus $W_n$ may become negative. Since our dynamics (1) are not built to describe agents in debt, we replace $W_n$ with one of three replacement mechanisms discussed in Section 4.1. We note that bankruptcy events where agents' losses exceed their current wealth are realistic and do occur, but in this paper we focus on modelling the dynamics of agents with positive wealth.

We also note that both, the non-stationary linear and super-linear models, exhibit **monopoly**, where the wealth fraction of the richest agent in a system of $N$ independent agents tends to 1 as time $n \to \infty$. This behaviour is well known for distributions with heavy tails (see e.g. Table 3.7 in [31]), which include the log-normal distribution in the linear case (4), and is only more pronounced in the super-linear model with heavier tails. We present related numerical results for the Gini coefficient and the top 1% wealth share in simulations, both tending to 1 in the long-time limit. While of course this extreme limit is not realistic currently, inequality measures are well known to increase since the 1980s (see summary in Section SI.2.5 in S1 Appendix). This is consistent with understanding current wealth distributions as transient behaviour of our model, which leads to monopoly if parameters remain unchanged over time. Of course we can only parametrise our model over the current range of wealth values, and in order to get more realistic forecasts for future wealth distributions, we would have to include also the lifetime and inheritance dynamics for agents and the role of external influences (such as war or other catastrophies). The simplified model we present here explains how current wealth distributions can arise naturally from generic initial conditions, and we discuss possible refinements for further study in Section 5.

## 3 Data analysis

Before moving on to the simulations of the non-linear Kesten process (3) we undertake some key empirical analysis to parametrise the model. We calculate returns on wealth, $R_n$, and the prefactor, $\alpha_n$, and make statistical fits on these variables. Although residual savings do not evolve with wealth as mentioned above, they are correlated with initial wealth values of an agent as part of their social status or fitness. To infer this dependence, we look at UK income and expenditure data for the year 2016 [32, 33].

### 3.1 Statistical properties of returns $R_n$

We rearrange (1) to find the ROR as

$$R_{n+1} = \frac{W_{n+1} - W_n - S_{n+1}}{W_n} \approx \frac{W_{n+1} - W_n}{W_n} \quad \text{for billionaires.} \tag{7}$$

For wealthy agents, wealth gain is to a large extent dominated by returns on wealth, so that $W_{n+1} - W_n \gg S_{n+1}$ and residual savings can typically be ignored. The ROR is then simply given by the wealth growth rate, which we will use to compute $R_n$ for billionaires, while we include residual savings to estimate ROR from survey data for other agents.

As mentioned previously, fairly recent work [6–8] has suggested an increasing wealth dependence on returns. We also find empirical evidence for this from WAS as summarised in Fig 2, and assume a simple power-law relationship as in (2) which is roughly consistent with the data. According to this we have

$$\mathbb{E}[R_{n+1}|W_n] = \mu W_n^{\gamma-1} , \quad \text{where} \quad \mu = \mathbb{E}[\alpha_{n+1}]. \tag{8}$$

We fit the power-law exponent $\gamma$ and the prefactor $\mu$ as shown in Fig 2, and also find evidence that returns are independent across time and the variance of returns is proportional to the square of the mean returns as wealth increases (see Fig 3),

$$\text{var}(R_{n+1}|W_n) \approx 0.57 \, \mathbb{E}[R_{n+1}|W_n]^2. \tag{9}$$

Such a quadratic scaling relationship of mean and variance is common in multiplicative processes, and consistent with our model assumption (2), as is explained in Section SI.1.1 in S1 Appendix.

Note that the apparent structure in percentile returns data in Fig 2 for individual years does not constitute reliable information in our view, since the variation of the points is artificially decreased due to our numerical procedure as explained in Section SI.2.4 in S1 Appendix. Viewing all years as a combined dataset, we find an increasing wealth dependence of RORs consistent with a simple power-law relationship, which also matches well with data for billionaires. In the next subsection we present a method to estimate a reasonable value of the power-law exponent $\gamma$ so that both, WAS and billionaire return data, can be modelled well with our assumption on returns (2).

### 3.2 Fitting $\alpha_n$

With (3) we have in analogy to (7)

$$\alpha_{n+1} = \frac{W_{n+1} - W_n - S_{n+1}}{W_n^{\gamma}} \approx \frac{W_{n+1} - W_n}{W_n^{\gamma}} \quad \text{for billionaires.} \tag{10}$$

As illustrated in Fig 4, we choose the power-law exponent $\gamma = 1.075$, such that the return data from the WAS and billionaires can be best explained with a single power law of the form (2). We fit the distribution of the $\alpha_n$ (which we assume to be i.i.d.) with a shifted and scaled non-central $t$-distribution (nct), i.e. we take

$$\alpha_n \sim \text{nct}(k, c, l, s).$$

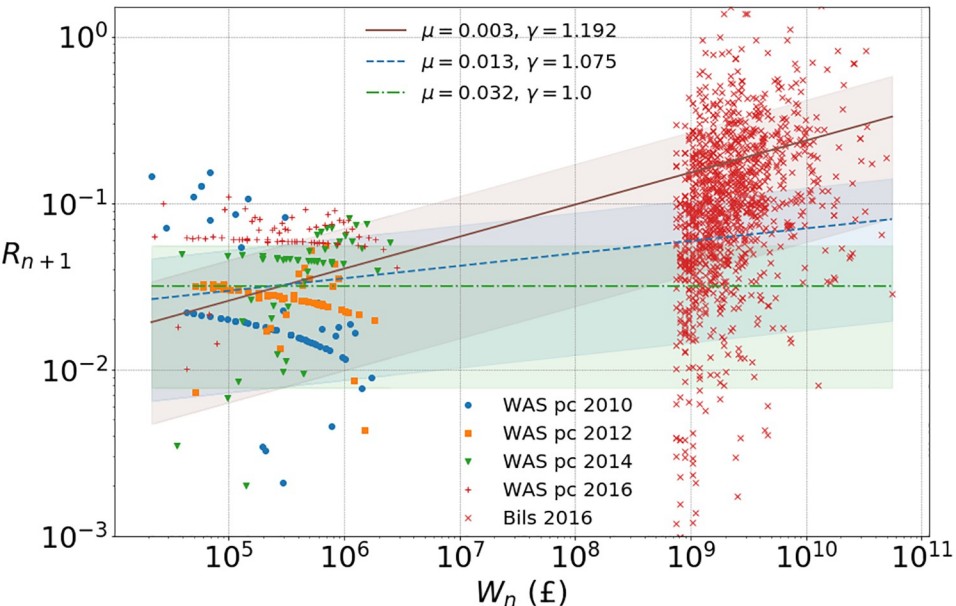

**Fig 2. Percentile ROR using WAS data [18] for the years 2010, 2012, 2014 and 2016, and ROR for individual billionaires for 2016 [19].** Power law fits according to (8) to the cluster of WAS ROR data combined over all four time periods, leads to $\mu \approx 0.003$, $\gamma \approx 1.192$ (with both parameters free) and to $\mu \approx 0.013$ with chosen $\gamma = 1.075$ (justified below in Fig 4). We also include $\gamma = 1$ for comparison, leading to $\mu \approx 0.032$, i.e. an average ROR of about 3%. Respective shaded regions are one standard deviation around the power fit means (15) as explained in Section SI.1.1 in S1 Appendix.

This distribution has four parameters: $k > 0$ represents the degrees of freedom controlling the heaviness of the tail, $c \in \mathbb{R}$ is the centrality that controls the skewness of the distribution, $l \in \mathbb{R}$ is the shift and $s > 0$ is the scale, see Section SI.2.6 in S1 Appendix for details.

We find that, while the bulk of the distributions of $\alpha_n$ agree well, the billionaire data lead to heavier tails than WAS data. Again, our method of extracting returns from WAS data leads to decreased fluctuations, and therefore we use the parameter values corresponding to billionaire data in simulations in Section 4.

### 3.3 Residual savings $S_n$

We recall that in our model (1) residual savings $S_n$ represent all contributions to wealth growth that are independent of the current wealth of an agent. They do not evolve with increasing

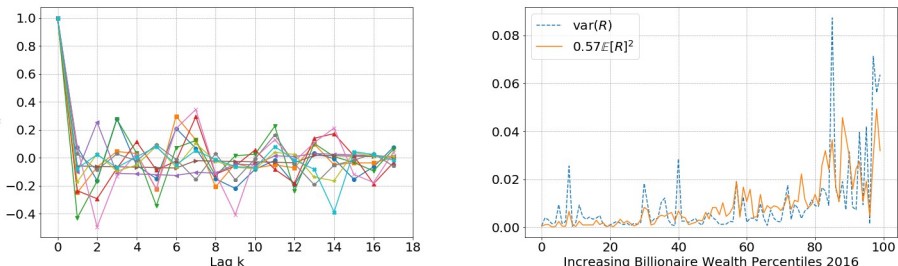

**Fig 3.** Left: autocorrelation of a sample of billionaire ROR indicating independence in returns. Right: average annual billionaire returns from 2008–2016 Forbes list [19], showing mean and variance relationship for increasing wealth percentiles as in (9).

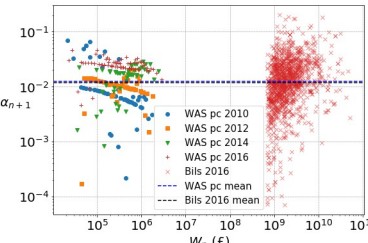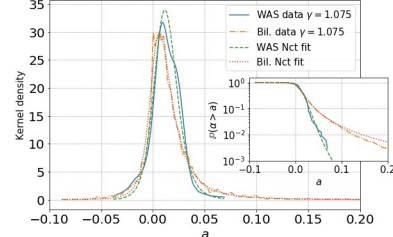

**Fig 4. Left:** $\alpha_{n+1}$ (10) for WAS data percentiles [18] for four time periods along with 2016 billionaire data plotted against wealth $W_n$. We choose $\gamma = 1.075$ so that the means of WAS and billionaire data essentially agree (dotted lines). Right: Kernel density of $\alpha_{n+1}$ for WAS data and 2016 billionaire data as seen in the left Figure. Inset: corresponding empirical tails $\mathbb{P}(\alpha_n > a)$ on logarithmic scale. Dotted green and red lines provide fits by the non-central $t$-distribution (nct) to WAS and billionaires with respective nct parameter fits $k \approx 6.03$, $c \approx 0.0573$, $l \approx -0.00575$, $s \approx 0.0112$ and $k \approx 2.01$, $c \approx 0.941$, $l \approx -0.00156$, $s \approx 0.0112$.

wealth and only contribute additive noise, which does not influence the long-time behaviour of the dynamics. However, we need to estimate residual savings and their correlation with (initial) wealth to run simulations, and in particular in order to extract empirical RORs from wealth data using (7), which determine the statistics of the crucial parameter $\alpha_n$. [34] presents evidence for recent years in the US, that income and salary are positively correlated with wealth.

We estimate residual savings by equivalised disposable income after expenditure for increasing deciles of median wealth using ONS data sources [32, 33]. Equivalised disposable income is household size adjusted income available for spending after tax and deductions, and by expenditure we summarise costs that do not contribute to wealth, such as buying food or paying rent. We fit the dependence on wealth $w$ with a logistic function

$$S(w) = \frac{\kappa_1}{1 + \kappa_2 w^{\kappa_3}} \quad \text{with parameters } \kappa_1, \kappa_2 > 0 \text{ and } \kappa_3 < 0. \tag{11}$$

This is illustrated in Fig 5, where we show data on equivalised disposable income, household expenditure and give the fitted parameter values for (11).

We used (11) as an estimate for additive contributions to wealth growth when calculating percentile returns in Fig 2, see Section SI.2.4 in S1 Appendix and in simulations in Section 4.2 as a function of initial wealth $w = W_0$. Note that the logistic fit levels off at $\kappa_1 = 10^6$ for large values of $w$ which is an arbitrary cap of $10^6$ GBP on wealth independent savings. For most rich

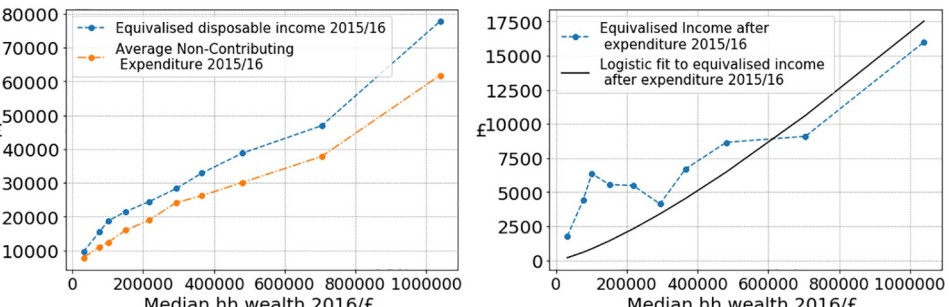

**Fig 5. Left:** Plot of equivalised disposable income and average household expenditure for 2015/16 against 2016 median wealth deciles. Right: Fit of the logistic function (11) to equivalised disposable income after expenditure, where we choose $\kappa_1 = 10^6$ and fit $\kappa_2 = 4.13 \cdot 10^9$ and $\kappa_3 = -1.308$. ONS data sources used can be found in [32, 33].

households, contributions to wealth growth significantly beyond this scale are in the form of wealth returns. It is important to note that none of our results are sensitive to the choice of parameters $\kappa_1$, $\kappa_2$ and $\kappa_3$, since savings only really play a role in parameter estimation or simulations on the scales shown in Fig 5.

## 4 Simulation results

For all simulations presented in this section we use i.i.d. $\alpha_n \sim \text{nct}(k, c, l, s)$ with parameters

$$k = 2.008 \ , \quad c = 0.941 \ , \quad l = -0.00156 \quad \text{and} \quad s = 0.0112 \ , \tag{12}$$

corresponding to data from individual billionaires which represent our best estimate of fluctuations for individual households for $\gamma = 1.075$, see Fig 4. We do, however, experiment with changing $\gamma$ values in which case we multiply the $\alpha_n$ by a positive constant to keep the mean at the same level. This is explained further in Section 4.1.

### 4.1 Generic initial conditions without residual savings

To investigate the general properties and dependence on initial conditions of our model over longer time horizons, we consider the following four different initial conditions each with mean 10000:

I.1. $W_0 = 10000$ (BLUE ●)

I.2. $W_0 \sim 5000 + \text{Exp}(1/5000)$ (ORANGE ■)

I.3. $W_0 \sim \text{Exp}(1/10000)$ (GREEN ▼)

I.4. $W_0 \sim \text{Pareto}(5000, 2)$ (RED +)

In other words, in **I.1** all agents start with initial wealth 10000, in **I.2** agents get 5000 plus an exponentially distributed random amount with mean 5000, in **I.3** initial wealth is drawn from an exponential with mean 10000 and in **I.4** it is Pareto distributed with scale parameter $x_m = 5000$ and exponent 2.

It is also possible in our simulations for the wealth $W_n(i)$ of an agent $i$ to become negative. In this case we choose one of the following replacements for $W_n(i)$:

R.1. replace with a proportion of the agent's previous positive wealth value $pW_{n-1}(i) > 0$ such that $p$ is uniformly chosen from $(0, 1]$

R.2. replace with the agent's previous positive wealth value $W_{n-1}(i) > 0$

R.3. replace with wealth $W_n(j) > 0$ of another uniformly chosen agent $j$

We can think of **R.1** as the agent losing a random proportion of wealth, **R.2** as no change in the agent's wealth and **R.3** as the agent being removed from the system and being replaced uniformly with another agent with positive wealth. We note that **R.3** is a simple approximation to resampling the agent's wealth from the current wealth distribution. We focus here on simulations with the more realistic compromise mechanism **R.1**. In Section SI.3.1 in S1 Appendix we will present simulation results for the more extreme replacement mechanisms **R.2** and **R.3** which lead to similar results, confirming that our model is not very sensitive on the choice of the replacement mechanism.

For each initial distribution we run the simulations iteratively using (3) for $N = 10^6$ independent agents and **zero residual savings** $S_n = 0$ with parameters in (12) and replacement mechanism **R.1**. We choose $S_n = 0$ for convenience in this section, to isolate the effect of the multiplicative dynamics which is dominant in generating the wealth distribution in this

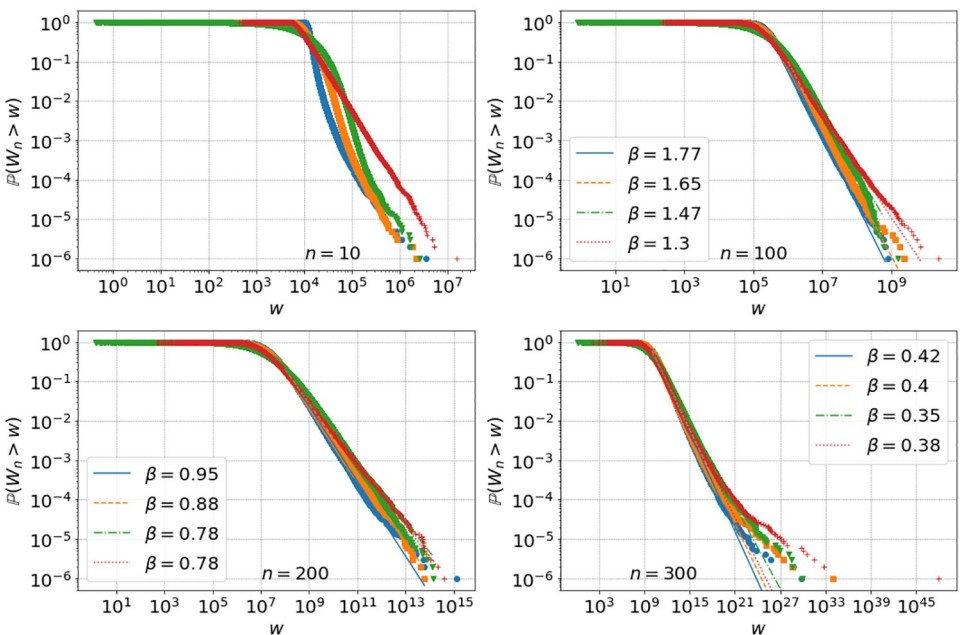

**Fig 6. Empirical tails for simulation (3) with $N = 10^6$ agents, residual savings $S_n = 0$, $\alpha_n \sim$ nct($k, c, l, s$) with $\gamma =$ 1.075, fitted parameters in (12), four initial conditions with colours and symbols as in I.1-I.4, and replacement mechanism R.1.** Power law fits show heavier tails with exponents $\beta$ decreasing with increasing times $n = 10, 100, 200$ and 300.

model, see Section SI.1.3 in S1 Appendix. Results for empirical tail distributions at times $n = 10, 100, 200$ and 300 are presented in Fig 6, using the colour code indicated in **I.1-I.4**. We also show standard inequality measures (see Section SI.2.5 in S1 Appendix for the definitions), the Gini coefficient $g$ and the top one percent income share $s_{0.01}$ for $\gamma = 1.075$ up to time $n = 300$ in the top left and right of Fig 7. We see that all initial conditions eventually lead to monopoly, and for intermediate times power-law tails emerge in the wealth distribution. Due to the crossover (6) to super-exponetial growth, a two-tailed structure emerges for large times and wealth values.

In Fig 8 we show for comparison empirical tails for $\gamma = 1.19$ with $\alpha_n \sim 0.23 \cdot$ nct($k, c, l, s$), and for $\gamma = 1$ with $\alpha_n \sim 2.5 \cdot$ nct($k, c, l, s$), so that average ROR values are well approximated with different fits for $\mu = \mathbb{E}[\alpha_{n+1}]$ (8) as shown in Fig 2. For $\gamma = 1$ we also compute the two inequality measures $g$ and $s_{0.01}$ up to $n = 400$, see bottom left and right of Fig 7 which shows the independence of initial conditions and slower progression towards monopoly. For the higher value of $\gamma = 1.19$ we see that the crossover sets in earlier at more realistic wealth values around $10^7$ with a two-tailed structure with quite realistic power-law tails (cf. Fig 1). For the linear model with $\gamma = 1$ we see no crossover and can fit the distribution for large times well by a log-normal distribution in accordance with (4). In this case there is also no noticeable difference between distributions originating from different initial conditions as we have seen in Fig 7. This is also illustrated in Fig 9, where we also see a clear dependence of final wealth values on initial conditions in the non-linear case with $\gamma > 1$.

## 4.2 Realistic initial conditions

In this section we simulate a realistic scenario for the UK, with $N = 23 \cdot 10^6$ households, initial conditions $W_0$ extracted from the UK wealth distribution in 2008, and with fixed (non-

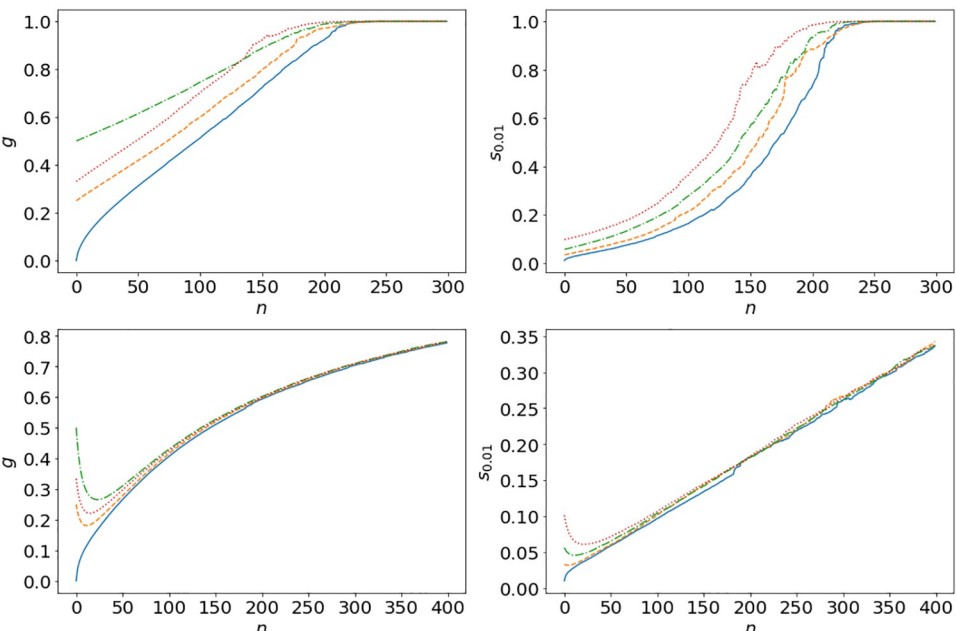

**Fig 7. Gini coefficient $g$ and top 1% wealth shares $s_{0.01}$ for simulation (3) with $N = 10^6$ agents, residual savings $S_n = 0$, $\alpha_n \sim \text{nct}(k, c, l, s)$ with fitted parameters in (12), $\gamma = 1.075$ for top left and right and $\alpha_n \sim 2.5 \cdot \text{nct}(k, c, l, s)$, $\gamma = 1$ for bottom left and right.** The four initial conditions **I.1** (full line), **I.2** (dashed), **I.3** (dash-dotted) and **I.4** (dotted) are used with replacement mechanism **R.1**.

random) residual savings $S_n \equiv S(W_0)$ as given in (11) of Section 3.3. Fig 10 shows the empirical tail of the resulting wealth distribution at times $n = 0, 2, 4, 6, 8, 10, 20$ and $50$, after simulating (3) with $S_n \equiv S(W_0)$, $\gamma = 1.075$, $\alpha_n \sim \text{nct}(k, c, l, s)$ with fitted parameters in (12) and replacement mechanism **R.1**. S5 Fig in Section SI.3.2 of S1 Appendix, shows empirical tails for the other two replacement mechanisms **R.2**, **R.3** which lead to very similar results. The number of agents $N$ is a rough estimate for the number of households in the UK with positive wealth in 2016. Time $n$ corresponds to the number of years after 2008, so for example $n = 8$ corresponds to 2016. Again we can see increasing inequality, see S6 Fig in Section SI.3.2 of S1 Appendix, with the decreasing power-law exponent $\beta$. In Fig 11 we show the corresponding average returns over time periods up to $n = 8$ for randomly selected agents, and find a very good correspondence with Fig 2 for empirical return data.

Comparing Figs 1 to 10 we see that the two-tailed structures differ slightly: While the heavier tail for billionaires with a power-law exponent of about $\beta = 1$ is shifting but well preserved, the stability of the lighter power-law tail for millionaires is not well represented in our simulation. This is because we deliberately chose a simple model assuming that average ROR follows a monotone power law with wealth. While this is largely consistent with data, the survey data for RORs show some plateau behaviour for millionaires clearly visible in Fig 2, which has also been suggested for other countries, see Fig 2 of [8]. This may be related to the changing wealth composition of the very rich [35].

## 5 Conclusions

The model defined by the iterative Eq (1) represents a generic evolution of household wealth, based on the well motivated assumption that wealth exchange between households does not play an important role. The particular form (3) of a non-linear Kesten process has been

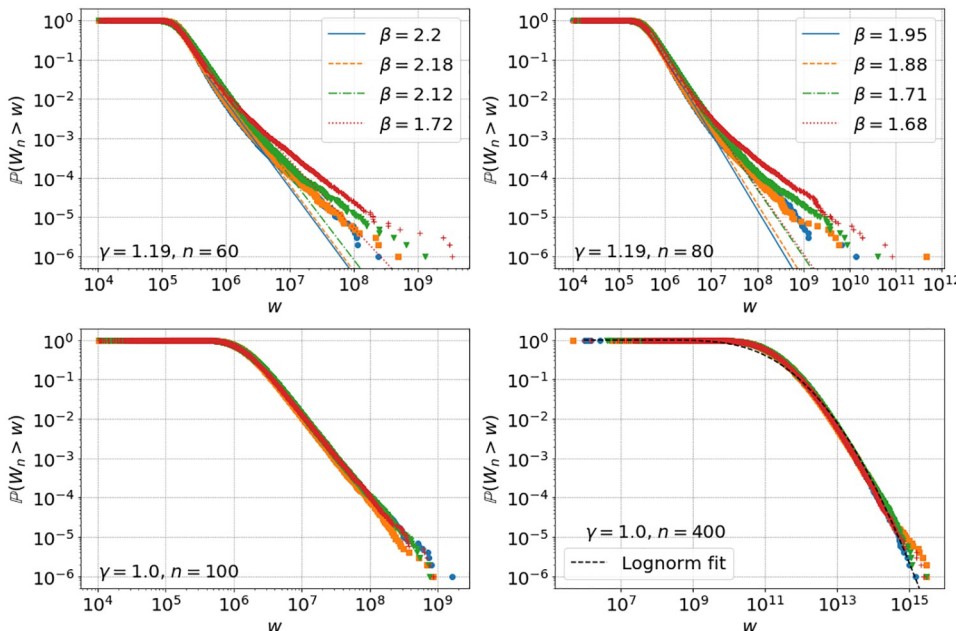

**Fig 8.** Top left and right: empirical tails for simulation (3) with $N = 10^6$ agents, residual savings $S_n = 0$, $\alpha_n \sim 0.23 \cdot \mathrm{nct}$ $(k, c, l, s)$ with fitted parameters (12) but with $\gamma = 1.19$ for the four initial conditions with colours and symbols as in **I.1**-**I.4**, replacement mechanism **R.1** and power law fits with exponents $\beta$. Bottom left and right: empirical tails for simulations as in top row, but with $\gamma = 1$, $S_n = 0$, $\alpha_n \sim 2.5 \cdot \mathrm{nct}(k, c, l, s)$, with lognormal fit at $n = 400$.

motivated by inferring empirically that RORs increase with household wealth, and that this relationship is consistent with a simple power law with exponent $\gamma$ as in (2), see also Fig 2. We want to stress that the qualitative results and main features of our model do not depend on this particular choice, which we have taken for simplicity and in order to study the effect of the non-linearity with a single parameter. We have seen from theory and simulations that the asymptotic dynamics of the model (3) and the resulting tail of the wealth distribution is dominated by the exponent $\gamma$. For the linear case with $\gamma = 1$ the RORs do not depend on wealth, and it is known that wealth grows asymptotically with a lognormal distribution (see Section SI.1.2 in S1 Appendix), which does not correspond to power-law tails seen in real data as in Fig 1. As demonstrated by our main results, the non-linear model with $\gamma > 1$ exhibits power-law tails

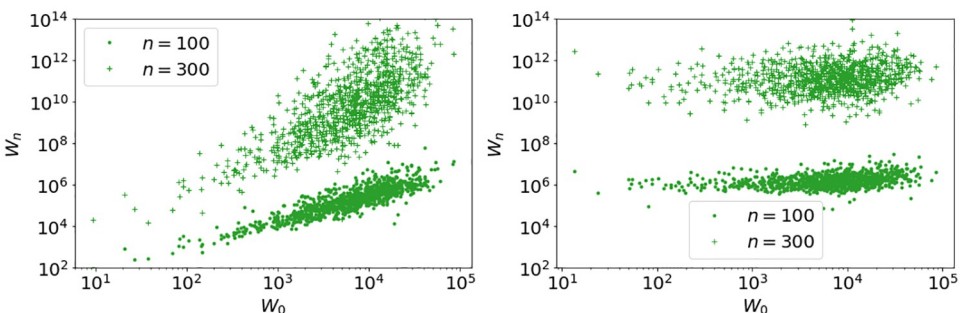

**Fig 9.** $W_n$ versus $W_0$ for 1000 randomly chosen agents for simulation (3) with $N = 10^6$ agents, residual savings $S_n = 0$ with fitted parameters in (12), left $\alpha_n \sim \mathrm{nct}(k, c, l, s)$, $\gamma = 1.075$ and right $\alpha_n \sim 2.5 \cdot \mathrm{nct}(k, c, l, s)$ and $\gamma = 1$. We use initial conditions **I.3** and replacement mechanism **R.1**. We see a clear dependence on initial conditions for $\gamma > 1$, and essentially no dependence for $\gamma = 1$.

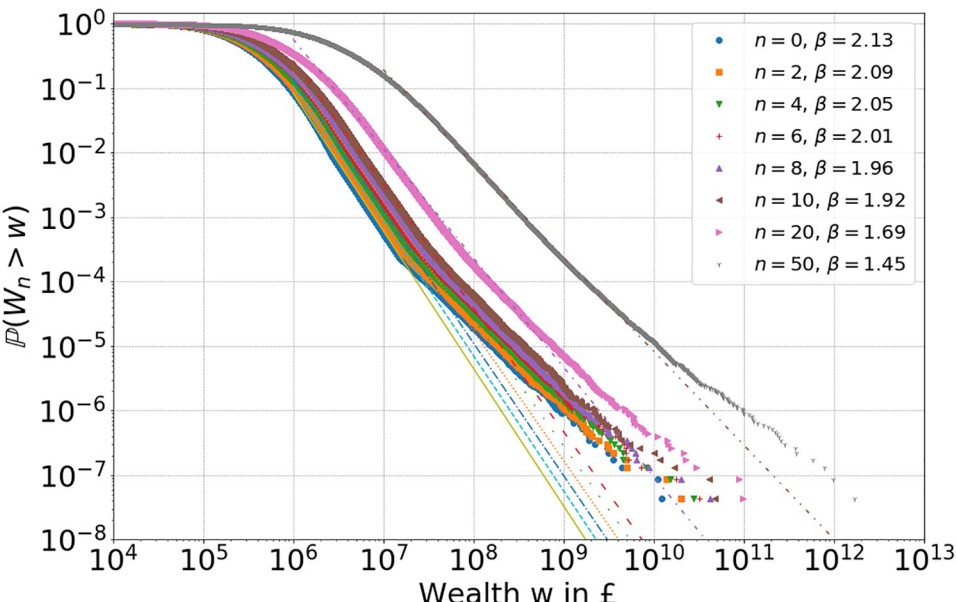

**Fig 10. Empirical tails for simulation (3) with $N = 23 \cdot 10^6$ agents, replacement mechanism R.1, $\gamma = 1.075$, fixed residual savings $S_n \equiv S(W_0)$ (11), $\alpha_n \sim$ nct($k, c, l, s$) with fitted parameters in (12) for 2008 initial conditions.** Fit values for a power-law tail exponent $\beta$ decrease from the initial value 2.13.

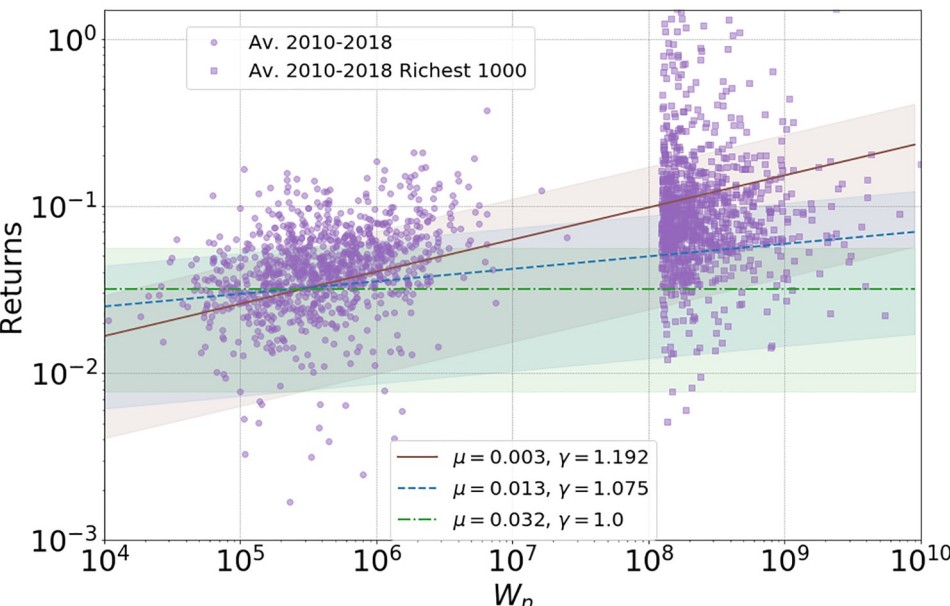

**Fig 11. Average return over 2010–2018 for randomly chosen agents against agents wealth in 2018.** The power fits (straight lines) for $\mathbb{E}[R_{n+1}|W_n] = \mu W_n^{\gamma-1}$, are the three fits to the real world data from Fig 2, along with one standard deviation error region.

from generic initial conditions, including even perfect equality or light tailed exponential distributions, see Section 4.1. It also leads to a two-tailed structure resulting from a crossover (6) to super-exponential growth for the richest households.

We now summarise the most important theoretical features and differences of the linear ($\gamma$ = 1) and the non-linear ($\gamma > 1$) non-stationary Kesten process (3):

- for all $\gamma \geq 1$, including the linear case, the model exhibits **monopoly**, i.e. for $N$ independent households the wealth fraction of the richest household increases with time and asymptotically approaches 1 (nevertheless, realistic levels of inequality can of course be achieved on intermediate timescales);

- the linear model is **ergodic**, in the sense that the asymptotic exponential growth rate of household wealth does not depend on the initial condition $W_0$. The latter only enters as a multiplicative factor and the model is **scale invariant**, i.e. wealth can be measured in units of $W_0$ in a dimensionless way;

- the non-linear model is **not ergodic**, i.e. the asymptotic exponential growth rate depends on $W_0$ and the early dynamics. It is also **not scale invariant**, and the non-linearity on the right hand side leads to a **critical scale** (6) where wealth gain per year can exceed current wealth, which is observed in data for the richest households.

Moreover, we would like to stress that our model is phenomenological and not built from first principles, since we simply assume an empirically motivated non-linear relationship between ROR and current wealth. Therefore the model lacks a natural scale invariance and the parameter $\alpha_n$ is not universal, but depends on the units of measurement (the currency) and will vary between different countries/economic areas. On the other hand, the non-linearity induces a crossover scale that can be a possible explanation for an apparent two-tailed structure in the data. This is an important aspect of our model which should be investigated further. While not present in data from the USA, the two-tailed structure has been observed [21, 22] for several countries which have a less liberal economic system and put more emphasis on social equality. Related political measures such as taxation then lead to a more even wealth distribution and a lighter power-law tail for rich households including millionaires, while the richest in society distribute their wealth globally and can escape such measures, leading to a heavier tail for billionaires.

Other interesting generalisations to make the model more realistic include dynamics for negative wealth, a realistic treatment of bankruptcy events and also household lifetime and fragmentation over longer time periods, or a household dependence of the parameter $\alpha_n$ reflecting variations in "fitness" to generate returns from investment. Also, mechanisms of household interaction possibly via a general redistribution or taxation procedure could be included and could lead to interesting effects on the dynamics similar to recent work in [15]. But the aim of this paper was to introduce a simple model, that can explain the main features of wealth distribution and dynamics, and how they can be explained by a non-linear wealth dependent rate of return.

## Supporting information

**S1 Appendix.**
(PDF)

## Acknowledgments

We would like to thank Alexander Karalis Isaac and Colm Connaughton for their helpful discussions on this work. S.G. would like to thank Technical University of Delft, where part of this research was carried out.

## Author Contributions

**Conceptualization:** Stefan Grosskinsky.

**Formal analysis:** Samuel Forbes.

**Funding acquisition:** Stefan Grosskinsky.

**Investigation:** Samuel Forbes.

**Methodology:** Stefan Grosskinsky.

**Project administration:** Stefan Grosskinsky.

**Validation:** Samuel Forbes.

**Visualization:** Samuel Forbes.

**Writing – original draft:** Samuel Forbes, Stefan Grosskinsky.

**Writing – review & editing:** Samuel Forbes, Stefan Grosskinsky.

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
