## [Decision Letter · Decision Letter 0]

9 May 2022

PONE-D-22-05886A Study of UK Household Wealth through Empirical Analysis and a Non-linear Kesten ProcessPLOS ONE

Dear Dr. Grosskinsky,

Thank you for submitting your manuscript to PLOS ONE. After careful consideration, we feel that it has merit but does not fully meet PLOS ONE’s publication criteria as it currently stands. Therefore, we invite you to submit a revised version of the manuscript that addresses the points raised during the review process.

Both referees suggest minor revisions and one of them is already in favour of acceptance. My suggestion is to carefully consider the points raised by the two referees and resubmit a revised version of the manuscript.

We look forward to receiving your revised manuscript.

Kind regards,

Enrico Scalas, Ph.D.

Academic Editor

PLOS ONE

Journal Requirements:

“We would like to thank Alexander Karalis Isaac and Colm Connaughton for their helpful discussions on this work. S.F. would like to acknowledge financial support from EPSRC through grant EP/L015374/1, S.G. would like to thank Technical University ofDelft, where part of this research was carried out.”

“S.F., EP/L015374/1, Engineering and Physical Sciences Research Council, https://epsrc.ukri.org/

Reviewers' comments:

Reviewer's Responses to Questions

**Comments to the Author**

1. Is the manuscript technically sound, and do the data support the conclusions?

Reviewer #1: Yes

Reviewer #2: Yes

2. Has the statistical analysis been performed appropriately and rigorously? 

Reviewer #1: Yes

Reviewer #2: Yes

3. Have the authors made all data underlying the findings in their manuscript fully available?

Reviewer #1: Yes

Reviewer #2: Yes

4. Is the manuscript presented in an intelligible fashion and written in standard English?

Reviewer #1: Yes

Reviewer #2: Yes

5. Review Comments to the Author

Reviewer #1: The paper investigates the wealth distribution of UK households through a detailed analysis of data

from wealth surveys and rich lists, and propose a non-linear Kesten process to model the dynamics of household wealth.

As a general comment, I think the paper makes an interesting contribution to the literature by analyzing household wealth. At the same time, I think the paper needs minor improvements before being published in this journal.

Specific comments

1) In the introduction section, at least a paragraph about agent-based models should be added.

2) I would suggest the authors add a new section “Data” to describe the empirical data used in the manuscript. Moreover, in this section, a Table reporting the main statistical properties of the data could be useful.

3) The authors could add a new section “conclusions”, where the main conclusions and the practical implication of the study are presented.

4) Please improve the quality of the figures, use not only different colors to plot the curves but also different line style or marker so that also in black and white the figures are readable.

Reviewer #2: this manuscript is overall well written. The methodology is coherent with their assumption that “the rate of return on wealth is increasing with wealth” in the extremely wealthy scenario, though Keynesian liquidity preference theory suggests the negative relationship between the rate of return and the amount of investment. We can also observe that Keynesian theory is partially relevant in Figure 2 as the downward slopes present. I think it is an interesting topic for readers to discuss.

6. PLOS authors have the option to publish the peer review history of their article (what does this mean?). If published, this will include your full peer review and any attached files.

Reviewer #1: No

Reviewer #2: **Yes: **Zheng NAN

---

## [Author Response · Author response to Decision Letter 0]

24 Jun 2022

Response to Referees:

Referee #1:

We thank the referee for the useful comments, which we have adopted as follows:

1) In the introduction section, at least a paragraph about agent-based models should be added.

We have added a short explanation of the agent-based approach in an appropriate place in the introduction on pages 1 and 2.

2) I would suggest the authors add a new section “Data” to describe the empirical data used in the manuscript. Moreover, in this section, a Table reporting the main statistical properties of the data could be useful.

We renamed Section 3 “Empirics” to “Data analysis”, since we think that it contains all the relevant statistical properties of the data that are relevant for the paper. The most important properties are summarized in Figures 1 and 2, and we were not sure how to add more relevant information in a table. After careful consideration we decided to leave those figures in place at the beginning of the paper, in order to motivate our choice of the model. Further technical details on data and the methodology to analyse them is given in the supplementary material in Section SI.2 Empirical analysis. 

3) The authors could add a new section “conclusions”, where the main conclusions and the practical implication of the study are presented.

We think that the content for the proposed section is present in 5 Discussion, which we renamed to 5 Conclusions.

4) Please improve the quality of the figures, use not only different colors to plot the curves but also different line style or marker so that also in black and white the figures are readable.

The quality of the figures has been improved as suggested. All Figures have been updated so they can now be differentiated in black and white using different line styles and markers.

 

Referee #2:

We thank the referee for carefully reading the manuscript and the useful comments, which we have adopted as follows:

3.1 Line 8 wealth/income shares. This expression using “/” is not clear sometimes. I suggest a verbal expression here. 

Implemented as: … as well as wealth or income shares,

3.2 Line 99 “a simple model that can also be analysed analytically. “ The term “be analysed analytically” seems pleonasm. Maybe, it can be replaced by be analysed using logical reasoning.

Changed to: … analysed mathematically. (which was the intended meaning)

3.3 Line 180 From (3) we rearrange to find the ROR. The resulting ROR is more relevant to the rearrangement from Equation (1).

Yes, we changed this as suggested.

3.4 L 89 Equation (1), L182 savings can typically be ignored, L213 We recall that in our model (1) savings S_n represent all contributions to wealth growth that are independent of the current wealth of an agent. Personally, there is a confusion about the meaning of the saving. From the expression in L213, we know that the saving is referred of as independent saving, a random valuable that doesn’t affect the growth of wealth and its mean value can be captured using a logistic function. However, in the convectional macroeconomic context, saving is equal to investment, which is obviously different from the meaning of saving in this manuscript. Maybe, a “residual saving” or “independent saving”, or the other word is a more accurate expression. Then, the residual saving makes sense in L182 that can be ignored comparing to the quantity of wealth. Consequently, please make it clear when giving Equation (1).

We have adopted this useful suggestion and now use the term “residual saving” throughout the manuscript.

3.5 Figure 6 Parameter β appeared once before Figure 6 as the parameter for Pareto distribution in I.4. What does β mean in Figure 6? Please avoid collision of notations, if β here suggests a power-law exponent.

We have removed β from Condition I.4 and only use it for exponents of power law fits in Figure 6 and later Figures.

---

## [Decision Letter · Decision Letter 1]

28 Jul 2022

A Study of UK Household Wealth through Empirical Analysis and a Non-linear Kesten Process

PONE-D-22-05886R1

Dear Dr. Grosskinsky,

We’re pleased to inform you that your manuscript has been judged scientifically suitable for publication and will be formally accepted for publication once it meets all outstanding technical requirements.

Kind regards,

Enrico Scalas, Ph.D.

Academic Editor

PLOS ONE

Additional Editor Comments (optional):

In the second round, the paper has been seen by only one of the original referees. However, I informally contacted the other referee who confirmed their positive opinion on this paper. I can see that the referee who analyzed the revised manuscript checked all the comments.

Reviewers' comments:

Reviewer's Responses to Questions

**Comments to the Author**

1. If the authors have adequately addressed your comments raised in a previous round of review and you feel that this manuscript is now acceptable for publication, you may indicate that here to bypass the “Comments to the Author” section, enter your conflict of interest statement in the “Confidential to Editor” section, and submit your "Accept" recommendation.

Reviewer #1: All comments have been addressed

2. Is the manuscript technically sound, and do the data support the conclusions?

Reviewer #1: Yes

3. Has the statistical analysis been performed appropriately and rigorously? 

Reviewer #1: Yes

4. Have the authors made all data underlying the findings in their manuscript fully available?

Reviewer #1: Yes

5. Is the manuscript presented in an intelligible fashion and written in standard English?

Reviewer #1: Yes

6. Review Comments to the Author

Reviewer #1: The paper has been improved following the referees’ suggestions, and now, according to me, it is ready to be published in this journal.

7. PLOS authors have the option to publish the peer review history of their article (what does this mean?). If published, this will include your full peer review and any attached files.

Reviewer #1: No

---

## [Editor Report · Acceptance letter]

1 Aug 2022

PONE-D-22-05886R1 

A study of UK household wealth through empirical analysis and a non-linear Kesten process 

Dear Dr. Grosskinsky:

I'm pleased to inform you that your manuscript has been deemed suitable for publication in PLOS ONE. Congratulations! Your manuscript is now with our production department. 

Kind regards, 

on behalf of

Professor Enrico Scalas 

Academic Editor

PLOS ONE